# The human fungal pathogen *Aspergillus fumigatus* can produce the highest known number of meiotic crossovers

Ben Auxier[1], Alfons J. M. Debets[1], Felicia Adelina Stanford[2], Johanna Rhodes[3], Frank M. Becker[1], Francisca Reyes Marquez[1], Reindert Nijland[4], Paul S. Dyer[2], Matthew C. Fisher[3], Joost van den Heuvel[1☯], Eveline Snelders[1☯]*

1 Laboratory of Genetics, Wageningen University; Wageningen, the Netherlands, 2 School of Life Sciences, University of Nottingham, Nottingham, United Kingdom, 3 MRC Centre for Global Infectious Disease Analysis, Imperial College London, London, United Kingdom, 4 Marine Animal Ecology, Wageningen University, Wageningen, the Netherlands

☯ These authors contributed equally to this work.

* eveline.snelders@wur.nl

**Data Availability Statement:** Data and code supporting the analysis used are available at https://github.com/BenAuxier/aspergillus_recombination, which is archived at https://doi.org/

## Abstract

Sexual reproduction involving meiosis is essential in most eukaryotes. This produces offspring with novel genotypes, both by segregation of parental chromosomes as well as crossovers between homologous chromosomes. A sexual cycle for the opportunistic human pathogenic fungus *Aspergillus fumigatus* is known, but the genetic consequences of meiosis have remained unknown. Among other Aspergilli, it is known that *A. flavus* has a moderately high recombination rate with an average of 4.2 crossovers per chromosome pair, whereas *A. nidulans* has in contrast a higher rate with 9.3 crossovers per chromosome pair. Here, we show in a cross between *A. fumigatus* strains that they produce an average of 29.9 crossovers per chromosome pair and large variation in total map length across additional strain crosses. This rate of crossovers per chromosome is more than twice that seen for any known organism, which we discuss in relation to other genetic model systems. We validate this high rate of crossovers through mapping of resistance to the laboratory antifungal acriflavine by using standing variation in an undescribed ABC efflux transporter. We then demonstrate that this rate of crossovers is sufficient to produce one of the common multidrug resistant haplotypes found in the *cyp51A* gene (TR$_{34}$/L98H) in crosses among parents harboring either of 2 nearby genetic variants, possibly explaining the early spread of such haplotypes. Our results suggest that genomic studies in this species should reassess common assumptions about linkage between genetic regions. The finding of an unparalleled crossover rate in *A. fumigatus* provides opportunities to understand why these rates are not generally higher in other eukaryotes.

10.5281/zenodo.8167717. Genome assemblies of parental strains AfIR964 and AfIR974, as well as sequence data of offspring are available under BioProject PRJNA768288. Sequence data for crosses NL1, UK1 and UK2 are available under BioProjects PRJNA947770, PRJNA947515 and PRJNA947514 respectively. Raw data underlying all figures is available from https://doi.org/10.5281/zenodo.816291.

**Funding:** This work was supported by the Nederlands Wetenschappelijk Organisatie (ALWGR.2017.010 to BA), The European Society of Clinical Microbiology and Infectious Diseases 2018 Research Grant (3184200058 to ES), and the Wellcome Trust (219551/Z/19/Z to JR, FAS, PSD, MCF). The funders had no role in study design, data collection and analysis, decision to publish, or preparation of the manuscript. The other authors received no specific funding for this work. MCF is a CIFAR Fellow in the Fungal Kingdom programme.

**Competing interests:** The authors have declared that no competing interests exist.

## Introduction

Fungi in the genus *Aspergillus* have long been a genetic model and primarily focused on *A. nidulans*. Studies in this species have unraveled the genetics of asexual spore production, as well as understanding of mitosis [1,2]. Aspergilli can be either heterothallic, requiring 2 opposite mating types for sexual reproduction or homothallic where a single strain can complete the sexual stage. Early genetic studies on *A. nidulans* found that introduced mutations on the same chromosome were often unlinked, indicating that this species has a high rate of recombination, high enough that the initial linkage maps required data from much rarer mitotic recombination [3,4].

While many species in this genus were historically thought to lack a sexual cycle, recent advances have uncovered a meiotic stage in an increasing fraction [5,6]. In these Aspergilli, during sexual reproduction after fertilization, fruiting bodies are formed, called cleistothecia, in which many dikaryotic cells develop with nuclei of both parents. Fusion of these nuclei results in many transient diploid meiocytes that enter meiosis followed by a post-meiotic mitotic division to produce 8 unordered spores per ascus [7]. Invasive aspergillosis, caused by the ascomycete fungus *Aspergillus fumigatus* is a serious life-threatening human disease. Clinical and environmental isolates are often resistant to antifungal azole treatments, developing from mutations either during clinical treatment or previous azole exposure in agricultural settings [8–10]. The sexual nature of *A. fumigatus* has been demonstrated in laboratory settings and signatures are visible in population genetic analyses [6,11,12].

Already the sexual cycle has been used to dissect antifungal resistance in clonally related strains [13] and recently in a backcrossed genetic design [14]. Sexual crosses have also been used to produce mating-compatible isogenic strains and identify further antifungal resistance [15]. However, a fundamental study for the impact of sexual events on genetic diversity for this ubiquitous mold has still not been performed. With growing interest in population biology and genetics [11,16–19], understanding the effect of sexual recombination is increasingly necessary to understand the emerging phenomenon of antifungal resistance in *A. fumigatus*. To meet this need, we constructed the first recombination map of *A. fumigatus*.

## Results and discussion

We used previously identified fertile strains AfIR964 and AfIR974 as parents in our cross [6,20]. Genome assembly of the 8 haploid chromosomes using short- and long-read data for each parent showed they were largely syntenic to both themselves as well as the reference Af293, therefore, lacking structural variation between them which could interfere with recombination (S1 Fig and S1 Table). From the AfIR964 x AfIR974 cross, we randomly isolated 195 haploid offspring from several cleistothecia and generated approximately 90X depth of short-read data from each. Mapped against the AfIR974 parent, we identified 14,113 high confidence variants based on quality and segregation criteria (S2 **Table**). These variants were unevenly spaced across the genome with a median inter-marker distance of 206 bp but a mean of 2,010 bp.

Using a naïve criterion of recombination as non-parental adjacent markers (i.e., A-B or B-A instead of A-A or B-B parental), we observed an average of 132.2 recombination events per offspring (mean 16.5 observed per chromosome, reflecting 33.0 chromosome bivalent per meiosis). After correcting for unseen double crossovers in homozygous regions using Haldane's mapping function, we infer an average of 148.4 crossovers spread across the 8 chromosomes, resulting in a surprisingly long genetic map length of 14,843 cM (Fig 1A and S3 Table). The mean recombinant fraction between adjacent markers was 1% (max 38%), indicating that our markers can provide an accurate genetic map (S2 Fig). This naïve criterion of a

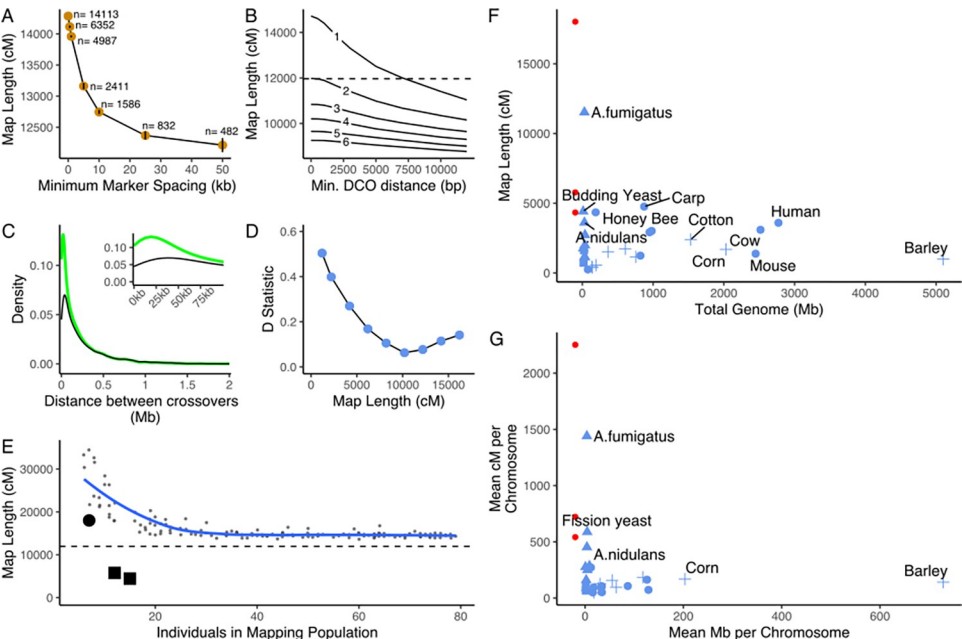

**Fig 1. Genetic map of *A. fumigatus*.** (**A**) Total map length after removing markers closer than specified interval (x-axis), lengths represent mean of 10 random samplings. Number of markers at specified spacing indicated near points. (**B**) Genetic map length after removing putative gene conversions based on length (x-axis). Criteria of minimum required markers is indicated in-line. (**C**) Distribution of DCO distances in raw data (green line) or after removing putative GC events (black line), inset shows zoom on first 100 kb distance. (**D**) Plot of D statistic comparing observed data to simulated genetic map lengths (see also S3 Fig). (**E**) Effect of subsampling population to smaller sizes shown in small gray dots, with loess line of best fit shown in blue. Additional mapping population NL1 shown by large black circle, and UK1 and UK2 populations shown by large black squares. Dotted line indicates genetic map after rarefaction at 20 cM marker spacing. (**F**) Comparison of the genetic map of *A. fumigatus* (blue triangle) to other species and genetic map sizes of additional *A. fumigatus* crosses in red circles. Red circles shifted left for visibility but have the same genome size as *A. fumigatus*. (**G**) Similar to F but shown per chromosome. Data for F+G derives from [23] with the addition of [5,24] and our data. Comparison to an alternate dataset is found in S4 Fig. Data underling this figure can be found at https://doi.org/10.5281/zenodo.8167717.

recombination event between homologues includes both crossovers as well as gene conversions that result from recombinational repair of meiotic double-strand breaks. Crossover recombination is reciprocal and results in the exchange over larger chromosomal intervals, whereas gene conversions are non-reciprocal exchanges over a relatively small interval termed gene conversions, and occur both in crossover and non-crossover events [21]. Since gene conversion events appear as 2 closely spaced crossover events, high-density genotyping can greatly inflate the genetic map [21]. Therefore, this distance is an overestimate and requires correction.

While in some fungi, it is possible to analyze individual meiotic tetrads, such techniques do not yet exist for *A. fumigatus*. This reduces the ability to distinguish gene conversion events based on their non-reciprocal nature. Thus, we used several analysis methods to remove gene conversions from our data, we first used a rarefaction of our markers. Since in general gene conversion tracts are short, typically 0.5 to 2 kb [22], high-density marker sets have increased power to detect them while fewer markers are necessary to detect crossover events, which affect chromosomal segments orders of magnitude longer [21]. Thus, increasing the gaps between markers, reducing the numbers of markers used, reduces the ability to detect gene conversions but less so for crossovers. Consistent with this, rarefaction of our markers reduced the map length asymptotically (Fig 1A). The reduced map lengths are not simply an artifact of

a reduced number of markers. For example, using a 10 kb spacing resulted in an average of 1,586 markers and a 12,745 ± 20 cM map, while at 50 kb spacing an average of 482 markers and 12,213 ± 102 cM—an approximately 75% reduction in markers only resulted in a 4.2% reduction in map length.

To test whether this high recombination rate was unique to the parents we used, an artifact of the laboratory environment or confounded by the high number of offspring analyzed compared to prior studies, we performed 3 additional crosses in addition to the original cross of environmental Irish strains AfIR964 and AfIR974 (S4 **Table**). In the Netherlands, we then also sequenced 7 offspring of an additional cross (NL1) between 2 Dutch environmental isolates, while in the United Kingdom, 16 offspring from a cross (UK1) involving one of the parents, AfIR974, of our original cross and 12 offspring of a separate cross (UK2) were sequenced independently (S4 **Table**). We then filtered the data in a similar manner to our original cross and determined the genetic map length. As these were noticeably smaller sample sizes than our original experiment, we also generated an expected effect of sample size on genetic map distance by random subsetting. The additional cross between Dutch isolates (NL1) resulted in a longer map of 18,013 cM, an increase that is consistent with our original map distance after correcting for smaller sample size (Fig 1E). However, the 2 crosses performed in the UK resulted in lower genetic distances (5,678 cM UK1 and 4,436 cM UK2), outside of the expectations for a sample of this size.

This very high recombination rate is consistent with alternate methods. First, post hoc criteria for removing gene conversions of a minimum number of consecutive markers or length supporting a double crossover (DCO) had differing effects on the resulting map length (Fig 1B). Increasing the minimum DCO length up to 12 kb resulted in an 11,037 cM map for our original cross. However, increasing the minimum number of consecutive markers for identification of a DCO had a stronger effect on map length, with a three-consecutive-marker criterion (i.e., ABBBA counting as a DCO) reducing the map to 10,839 cM (Fig 1B). This stronger effect of consecutive marker number is likely due to uneven marker distribution resulting in portions of the genome with few markers, where some true DCOs may only be represented by a few variants. Secondly, as a further map length estimation method, we simulated differing map lengths based on uniformly distributed crossovers across the genome. The related *A. nidulans* has been shown to lack crossover interference, with DCO distances producing a Gamma distribution with a shape parameter, $v$, of 1 [25,26]. We found that lengths between 11,000 and 13,000 cM best fit the data, minimizing the D statistic of these DCO distances between observed and simulated data (Fig 1D and additional map lengths in S3 Fig).

Therefore, based on our original cross, we estimate the genetic map length of *A. fumigatus* is between 11,000 to 13,000 cM, and we use the 11,966 cM estimate resulting from 50 cM marker rarefaction. This map length equates to 29.9 crossovers per chromosome pair during meiosis or 15 crossovers per chromosome per haploid offspring. To our knowledge this map length, resulting from such a high number of crossovers per chromosome, is the longest estimated for any eukaryotic organism (Figs 1F and S4A) even after correcting for chromosome length (Figs 1G and S4B). These values are also higher than a recent analysis of the fission yeast *Schizosaccharomyces pombe* [27]. Further support for this exceptionally high recombination rate comes from a previous study, using different parents that showed >5% recombination between 2 spore-color genes, *alb1* and *abr2*, spaced 8.3 kb apart (i.e., >0.6 cM/kb) [20]. However, the high rate of recombination that we observe in *A. fumigatus* does not appear to be widespread across *Aspergillus* species. High-density whole genomic genetic maps for *A. nidulans* indicate a total genetic length of 3,705 cM, while crosses in *A. flavus* indicate between 1,504 and 2,001 cM, respectively [5,24]. Clearly, the impact of this recombination rate on the population genetics of *A. fumigatus* will depend on the frequency of sexual versus asexual

clonal reproduction. Using the $LD_{50}$ metric—the average distance at which population-level genetic linkage decays to half of its maximum value (see Nieuwenhuis and James for a complete explanation [28])—across *A. fumigatus*, an $LD_{50} < 10$ bp has been observed [11]. Such a rapid decay of genetic linkage (in many other species, $LD_{50}$ values are often $>1$ kb [29,30]) indicates that sexual recombination in *A. fumigatus* field populations occurs with sufficient frequency to unlink clonally derived haplotypes. Further supporting the frequency of sex in the environment is the narrow region of diversity, approximately 250 bp, surrounding regions under strong balancing selection like the *het* or *mat* loci, which have maintained divergent alleles for several millions of years [31].

Our results indicate that gene conversions are responsible for a minority of genetic recombination in *A. fumigatus*. In budding yeast, slightly more than half of DSBs are converted to crossover events [21], while for the model plant *Arabidopsis thaliana*, it appears that almost all DSBs are resolved through non-crossover gene conversions [32]. Without sequenced tetrads, we cannot confidently identify individual gene conversion events, but we can assess the relative abundance. The removal of gene conversion events based on marker rarefaction reduced the genetic map by 15% (Fig 1A) and conservative thresholds reduced this length by 30%. This relatively low influence of gene conversions may indicate that, like budding yeast, most DSBs are converted to crossovers events. Clearly, future work including tetrad analysis is necessary to fully understand this.

As suspected based on the relationship to *A. nidulans*, we find no evidence supporting crossover interference—the phenomenon where the location of 1 crossover affects the location of adjacent crossovers [26]—although recombination is not uniform across the physical chromosome (Fig 2B). We base this conclusion on 3 observations. First, as a direct measurement of interference [33], comparing observed and expected double crossover rates across adjacent sets of 3 markers we find a coefficient of coincidence of approximately 1 (Fig 2C). Secondly, the distribution of crossovers per chromosome is very similar to the Poisson distribution expected when assuming an absence of crossover interference (Fig 2D) [26]. Finally, in the absence of interference the shape parameter of the interevent Gamma distribution, *v*, is expected to be 1, with values greater than 1 indicating interference (most species $5 < v < 30$) [26]. Our observed *v*, 0.74, (Fig 1C) does not indicate crossover interference, although whether this is due to uniform DSB patterning or rather the crossover maturation process cannot be disentangled with this metric [34]. Mechanistically, the lack of crossover interference in *A. nidulans* has been connected to the coincident lack of a meiotic synaptonemal complex (SC), similar to the fission yeast *Schizosaccharomyces pombe* [25,35], which also lacks crossover interference [27]. We thus expect that *A. fumigatus* also physically lacks a meiotic SC. However, physical evidence for the absence of an SC should be investigated. Bioinformatic analyses for presence/absence of SC proteins are not feasible, due to the high rate of evolution of these proteins [36], as the closest related species with a characterized SC is *Sordaria macrospora* [37]. Interestingly, the number of crossovers per individual in our data (min. 80; max. 300) was over-dispersed compared to the expected Poisson distribution (Fig 2E). This positive correlation between the number of crossovers on an individual chromosome and the total number of crossovers occurring within a meiotic nucleus is consistent with a recently recognized phenomenon (Fig 2F) [27,38,39]. The variation in numbers of crossovers was even more pronounced in some of our additional crosses, with individuals from the UK2 cross having between 20 and 128 crossovers, from the UK1 cross between 112 and 210, and the NL1 cross 96 to 218.

As validation of this genetic map length, we tested the translation to genetic mapping. Fortuitously, we observed phenotypic variation between the parental isolates AfIR964 and AfIR974 for resistance to acriflavine (Fig 3A), an antifungal with a rich history in fungal

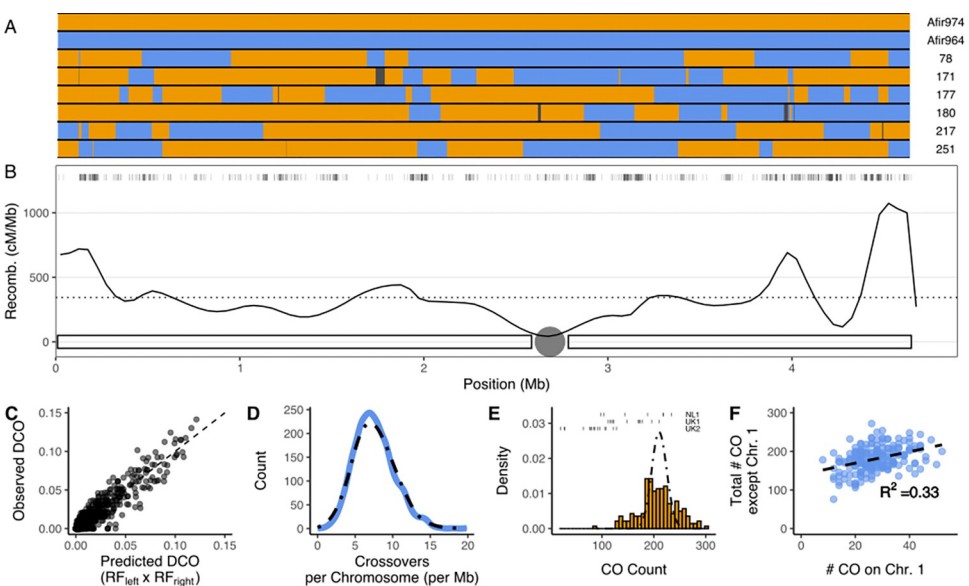

**Fig 2. Crossover distribution in *A. fumigatus* shows no sign of interference. (A)** Genotype of parents and 5 representative offspring for Chromosome 1. Blue indicates AfIR964 genotype, orange indicates AfIR974. Black regions indicate regions removed from gene conversion criteria. **(B)** Recombination rate across Chromosome 1. Solid line indicates recombination rate across windows of 50 kb, dotted line indicates mean recombination across Chromosome 1. Tick marks above indicate markers used for mapping, and chromosome with centromere is indicated below. Plots for all chromosomes are found in S5 Fig. **(C)** Correlation between predicted DCO based on recombinant fraction (RF) in adjacent intervals and observed DCO across the same interval in the 195 offspring. Diagonal dotted line indicates a 1:1 relationship, a coefficient of correlation of 1. **(D)** Histogram of normalized number of crossovers per Mb of chromosome, compared to Poisson distribution of same mean (dotted line). **(E)** Number of crossovers per individual in the dataset (orange bars) compared to the Poisson distribution of the same mean (dotted line). Total numbers of crossovers per individual for 3 additional crosses are indicates as tick marks above. **(F)** Scatterplot showing for each offspring the number of crossovers on Chromosome 1 against the total number of crossovers (excluding Chromosome 1). The dotted lines indicates the line of best fit. Data underling this figure can be found at https://doi.org/10.5281/zenodo.8167717.

genetics [40]. QTL analysis of growth on 50 μg/mL acriflavine identified a single locus on chromosome 6 between positions 657 kb and 675 kb, the resolution of which was restricted by the limited number of variants, rather than recombination (Fig 3B). Similar mapping of a mendelian trait in the haploid offspring of *Armillaria ostoyae*, a species with a lower recombination rate (1,007 cM) resulted in a window of 87 kb [41]. This exceptionally high resolution of the genetic mapping in *A. fumigatus*, an 18-kb window, highlights the power of genetics in this species to identify novel variation. The 2 variants in this region fall within the coding sequence of AFUA_6G03080 (Fig 3B). This gene encodes an undescribed ABC efflux transporter in the family that also contains Multi Drug Resistance 1 (*mdr1*) [42]. The difference in LOD scores between these 2 variants, from recombinant offspring across this distance of 410 bp, allows for immediate fine mapping of a presumed causal variant (Phe to Cys). Previously, genetic methods have been used to identify antifungal resistance mechanisms in *A. fumigatus* using clonally related isolates [13], backcrossed isolates have been used to identify genetic variation leading to azole resistance [15], and recently established tools like bulk segregant analysis been applied to understand antifungal resistance in this species [14]. Here, we show that variants between wild-type *A. fumigatus* isolates can be easily mapped even for genes without known mutant phenotypes. The recent genetic identification of 5 *het* loci using the same parents as described here demonstrates the power of genetics in a species with such a high recombination rate [31].

The high rate of recombination observed may further explain observed patterns of antifungal resistance in *A. fumigatus*, a major concern in the clinical setting [9]. Azole-resistant

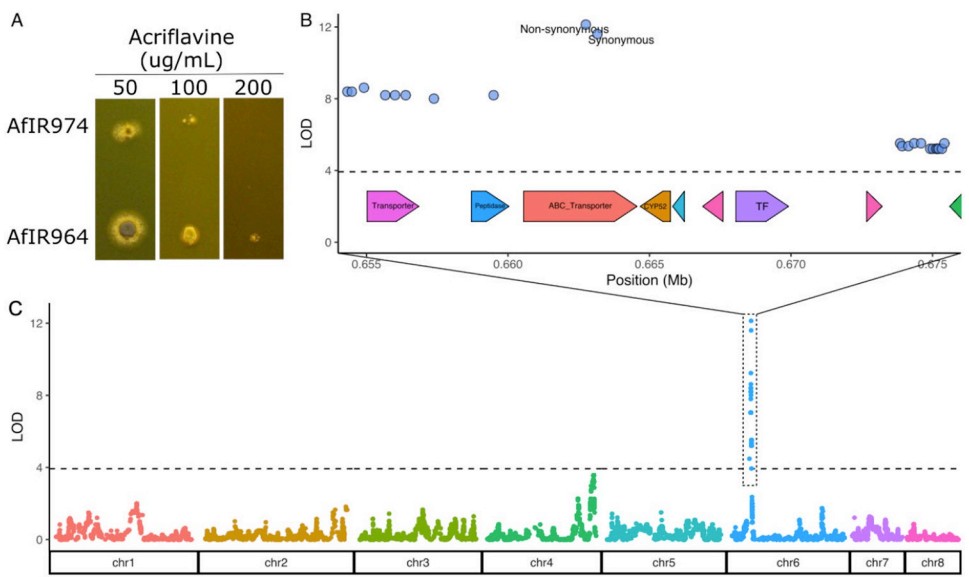

**Fig 3. Validation of *A. fumigatus* recombination rate through mapping of acriflavine resistance. (A)** Phenotypes of both parental strains grown at increasing concentrations of acriflavine after 2 days incubation. **(B)** Closeup of the significant QTL window. Gene models shown below the dotted line indicate location of genes in the AfIR964 parent. **(C)** Genome-wide significance values for acriflavine resistance in the cross between AfIR964 and AfIR974. Horizontal dotted line indicates the permutation-based significance threshold with α = 0.05. Data underling this figure can be found at https://doi.org/10.5281/zenodo.8167717.

environmental isolates commonly contain haplotypes of 2 to 3 variants in the *cyp*51A gene involved in ergosterol synthesis. These resistant haplotypes are composed of a tandem repeat (often 34 or 46 bp) in the promoter element (TR) combined with at least 1 non-synonymous polymorphism in the coding region (e.g., $TR_{34}$/L98H or $TR_{46}$/Y121F/T289A). There are strong epistatic effects, as the effects are non-additive between specific combinations of non-synonymous coding polymorphisms and promoter mutations causing increased antifungal resistance [43,44]. Curiously, it has been noted that in other fungal species resistance to the same azoles is instead generally due to single mutations in either the promoter or the coding sequence and not by a strong positive epistatic interaction of both, as found in *A. fumigatus* [10]. The *cyp51A* haplotypes in *A. fumigatus* have been hypothesized to arise from mutation/tandem duplication at 1 position and then an additional mutation/tandem duplication at the second position [10]. Our data here suggests an alternative explanation of each mutation arising in independent strains, and then being united through recombination, which is considered an evolutionary benefit of sexual recombination [45,46].

As the original parental strains had no variation either within or near the *cyp51A* gene, any recombination in this region would not be observed, although it has been noticed at a population level [17]. However, the 0.422 cM/kb recombination rate predicts that if one parent had the $TR_{34}$ variant, and the other had the L98H variant (650 bp apart), then 0.27% of offspring would be recombinant (0.14% of offspring would have both resistance variants, 0.14% with neither, and the remaining 99.83% having one or the other variant) (Fig 4A). Recombination in the $TR_{46}$ haplotype is expected to be slightly higher (as $TR_{46}$ and Y121F are spaced 719 bp apart). Since a single fruiting body produces >10,000 spores [20], recombinants within *cyp51A* gene are therefore expected in each sexual event.

To validate this predicted intragenic *cyp51A* recombination, we crossed previously generated single mutants [47] with AfIR974, generating sexually fertile offspring with either L98H

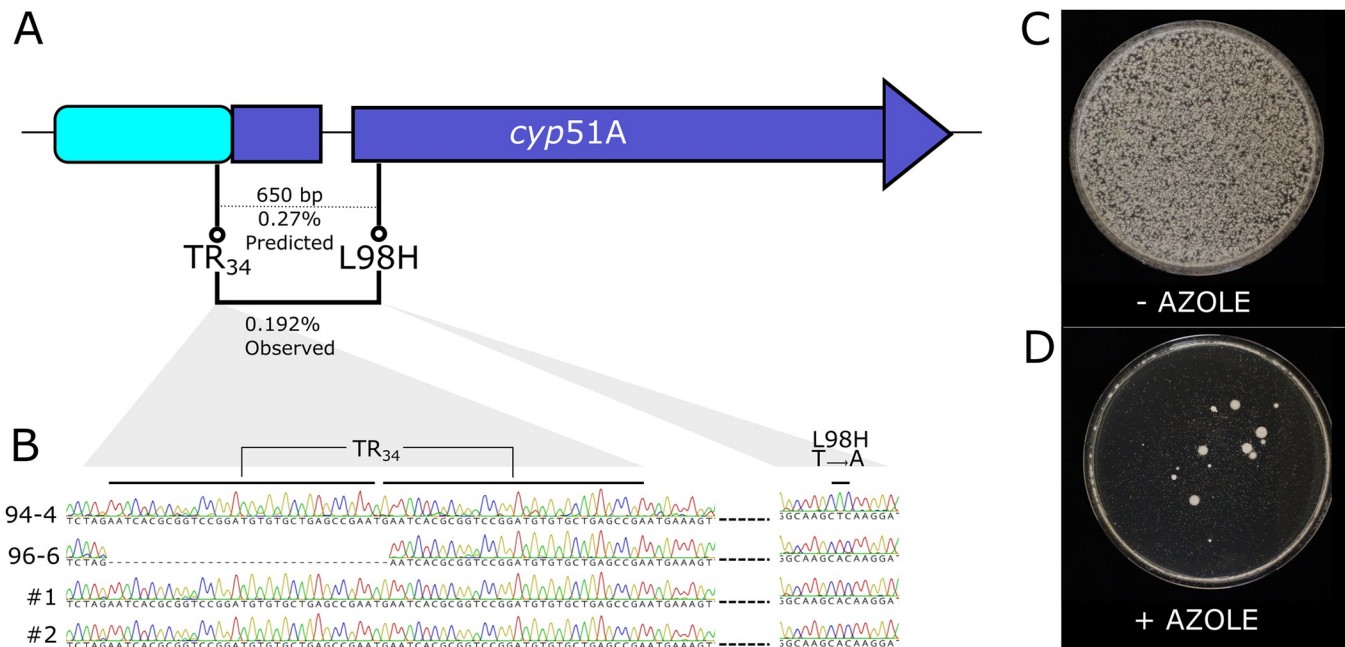

**Fig 4. Sexual recombination consistently produces recombinant *cyp*51A haplotypes. (A)** Diagram indicating distances and inferred recombination rates between positions in the *cyp*51A gene, leading to commonly encountered TR$_{34}$ and TR$_{46}$ azole resistance haplotypes. Base pair distances are indicated above dotted line, predicted recombination rates indicated below. Observed recombination rate from B indicated with solid line connecting TR$_{34}$ and L98H. **(B)** Chromatograms of *cyp*51A sequence showing TR$_{34}$ region and L98H region from parental strains with either variant, as well as 2 randomly selected azole resistant offspring. **(C)** Ascospores of a single sexual fruiting body on nonselective media. **(D)** As above, but with addition of 10 µg/mL itraconazole, which is above the tolerance of either single mutant strain. Large colonies indicate resistant offspring. Nonrecombinant offspring are visible as scattered small dots. Note that plate for Fig 4C was incubated 16 h less than D to prevent overgrowth of abundant colonies. Images of contents of all cleistothecia used are found in S6 Fig.

or TR$_{34}$ variants that each confer intermediate levels of azole resistance (4 µg/mL itraconazole). After crossing these, we then selected recombinants based on their positive epistatic effect by incubation on 10 µg/mL itraconazole (Fig 4C and 4D). These highly tolerant offspring were produced at a rate of 0.096% (mean number resistant colonies 18, mean total ascospores 18,708) and since the double wild-type recombinants are not scored, we infer that 0.192% of offspring are recombinant. To confirm that these highly resistant offspring did not arise from de novo mutations, we analyzed an additional set of phenotyping, which were produced at a similar rate as before of 0.098%, selecting 37 resistant colonies, of which 34 could be subcultured on azole-containing media. Sanger sequencing confirmed that all 34 were recombinant, with both the TR$_{34}$ duplication and the L98H variant (Fig 4B).

Fungi are known to generally have higher recombination rates compared to other eukaryotes [48], but our results provide an even higher level. Across eukaryotes meiosis usually produces a handful of crossovers per chromosome [49], and an observed negative correlation between population size and recombination rate may indicate that excess crossovers are detrimental [50]. The detrimental side of high recombination can result from the breaking up of beneficial mutations [45]. An increased number of crossovers like we observe has been suggested as a compensation to facilitate proper meiotic chromosome segregation in the absence of an SC [25,35]. It has become clear that the relationship between the SC and interference is not a simple cause and effect, since it may be that interfering crossovers trigger SC formation, not vice versa [51], and that the requirement for 1 crossover per chromosome might occur independently of other interference mechanisms [34]. Additionally, in *Arabidopsis*, deletion of several meiosis genes can greatly increase the number of non-interfering crossovers [52].

However, it also seems that the absence of SC-related proteins, the situation expected in *A. fumigatus*, leads to an increase in non-viable aneuploid progeny [53,54]. If this high crossover rate is simply a compensation for the lacking SC, it is unclear why the rate in *A. fumigatus* can be so much higher than that found in *A. nidulans* and *S. pombe*, which also lack an SC.

While we observed variation in recombination rate between the crosses we produced, the source of this variation remains unclear. Although these crosses were performed in different laboratories, conditions were similar during the stage where meiosis would have occurred. One of the additional crosses involved the use of AfIR974, one of the parents of the original cross, yet the recombination rate was reduced by approximately half. As such, it is unclear if the causes of this are due to predominantly genetic or environmental factors and will require further investigation. Identifying the range of recombination rates encountered in natural settings would be useful; however, recovery of sexual progeny from decaying vegetation is technically difficult [55], and experiments with large sterile compost heaps are impractical.

Our data shows that a physical limitation of crossover number does not explain the general low crossover rate across eukaryotes. Theoretically, only under a narrow range of parameters a constantly changing environment can select for higher recombination rates [56]. What ecological or evolutionary factors differentiate *A. fumigatus* from other *Aspergillus* species, with a lower recombination rate, remains unclear. However, an increased rate of recombination would facilitate the rapid generation of complex multi-azole-resistant haplotypes, which were detected in *A. fumigatus* before their occurrence in other fungi exposed to similar azole fungicide pressures [10]. As detection of *A. fumigatus* sexual structures indicates meiosis is occurring in azole-contaminated bulb waste heaps [55], our findings can explain the rapid production of the particular combinations observed of azole resistant *cyp51A* haplotypes [57,58]. The frequency of these epistatic haplotypes can then rapidly increase due to the selection pressure imposed in environments with azole fungicide residues [59]. This high recombination rate also affects the interpretation of population-level genome scans as it practically eliminates linkage between genes/markers [11]. Understanding the adaptive advantage, or lack thereof, of the unparalleled rate of recombination that we describe in this pathogenic mold requires further enquiry.

## Materials and methods

### Sexual crossing

Strains were crossed by plating complementary MAT1-1 and MAT1-2 strains on Oatmeal Agar 72.5 g/L Difco Chemical, or made in-house [14] followed by incubation at 30˚C for 4 to 6 weeks. Ascospores were harvested by manually isolating mature cleistothecia, and asexual conidia were removed by rolling in water agar. After crushing cleistothecia in water, the spores were heatshocked for 60 min at 70˚C to kill remaining asexual conidia and hyphae. Aliquots of this spore suspension were then plated on Minimal Media [60] with 0.1% Triton X-100 to restrict colony size. Isolated single-spore colonies were transferred to MEA slants (30 g/L Malt Extract; 1 mg/L CuSO$_4$) and incubated at 37˚C.

### DNA isolation and sequencing

For the crosses AfIR964 x AfIR974 and 88C19 X 46A23, DNA was extracted using a modified phenol-chloroform extraction method [61] from 24-hour-old mycelial mats from both parents and all offspring. Conidia were added to 2 mL of liquid Malt Extract media (30 g/L Malt Extract) and incubated for 48 h at 37˚C. The resulting mycelial mat was removed to a microcentrifuge tube, to which five to six 2 mm glass beads were added. This tube was then frozen in liquid nitrogen and homogenized with an Ivoclar Vivadent Silamat S6 homogenizer for 10 s at

4,500 rpm, twice. To this mycelial powder, 530 mL of Breaking Buffer (2% Triton X-100, 1% SDS, 10 mM Tris-HCl (pH 8.0), 1 mM EDTA (pH 8.0)) and 20 mL of 20 mg/mL Proteinase K was added. After vortexing to mix, the samples were incubated for 1.5 h at 56˚C. Following this, 550 μL of 24:24:1 Phenol:Chloroform:Isoamyl Alcohol was added, and samples were gently mixed on a rotary shaker for 10 min. Following this, the layers were separated using a table-top centrifuge at 14,000 rpm for 15 min. The extraction step was repeated on the aqueous layer. DNA was precipitated from the aqueous layer using an equal volume of isopropanol and washed using first 96% ethanol, followed by a wash with 70% ethanol. The resulting pellet was dried at room temperature and resuspended in 100 μL of warm TE buffer (10 mM Tris-HCl, 1 mM EDTA (pH 8.0)) and 1 mL of RNAse I (Promega) was added. This purified DNA was incubated for 1 h at 37˚C to digest RNA, and then frozen for storage. Meanwhile, for the crosses AfIR974 x 47–55 and C78 x 47–55 DNA were extracted as described [62].

For each offspring and parental strain from the crosses AfIR964 x AfIR974 and 88C19 X 46A23, an aliquot of DNA was processed for Illumina 150 bp paired-end sequencing (Novogene). DNA from offspring from crosses of AfIR974 x 47–55 and C78 x 47–55 were processed as previously described [62]. For each of the parental strains, an additional DNA aliquot was then processed for the Oxford Nanopore platform by first using the Circulomics SRE kit according to manufacturer's instruction to remove short DNA fragments, followed by the SQK-LSK-109 Genomic DNA by Ligation kit. Libraries for both parents were sequenced to approximately 100× depth using an Oxford Nanopore R9.4.1 flowcell (S1 **Table**). Basecalling was performed with Guppy software (3.2.8) with high accuracy settings.

## Genome assembly

A hybrid assembly strategy was followed, merging an assembly using the minimap2 (-x ava-ont, 2.17-r954)/miniasm (0.3-r179) and canu (v.2.1.1, trim-assemble genomeSize = 30m error-Rate = 0.05 -nanopore-raw)/racon (v1.3.1, -m 8 -x -6 -g -8 -w 500) pipelines [63–66]. For the latter, bwa-mem mapping was used (v0.7.15, -x ont2d) [67]. Both these assemblies were followed by pilon (v1.23) polishing before merging for which bwa-mem was used for mapping after which mapping was sorted and filtered (-q 20) using samtools (v1.9) [68,69]. After merging these assemblies, another round of racon correction and pilon polishing assured high contingency and accuracy. To test for genome completeness, BUSCO (v3.1.0) scores were calculated using the eurotiomycetes_odb9 database (—mode genome—species aspergillus_fumigatus) [70]. The final genome was annotated with augustus (v2.5.5,—gff3 = on—extrinsicCfgFile = extrinsic.E.cfg) using a hints file that was produced by using blat (v36.0, -t = dna -q = rna -minIdentity = 90 -minMatch = 1 -minScore = 10) to search the Af293 reference genome coding sequences against the assembled genome [71,72].

## Alignment, variant calling

Raw Illumina reads from both parents were aligned to the AfIR974 genome with bwa mem (v0.7.15), and subsequently filtered using samtools view (v1.9, -q 20), after which the resulting bam file was sorted. Reads groups were added using gatk AddOrReplaceReadGroups (v4.1.6.0) [73]. Resulting bam files were used to call variants using freebayes (v0.9.21) [74]. This raw VCF file was then annotated for effects of the variants using snpEff (v4.3t) using the above described augustus annotation.

## Offspring and variant filtering

We performed a conservative variant filtering using vcfR (v1.12.0) [75]. This involved first filtering the raw vcf for read frequency (>0.95 between parents) and coverage (20 < coverage <

150 for both parents) to identify loci that were differentiated between parents. This allowed filtering for individuals that were either not offspring of the 2 parents, exactly similar to one of the parents or showed heterozygous allele frequencies. This filtering removed 52 individuals that were parental, indicating significant conidial survival despite 60 min incubation in a heating block at 70°C. This contradicts previous reports of conidial death at incubation at 70°C [20], likely due to the difference in heat transfer between a heating block with air gaps and a more effective water bath. We then removed loci that often showed a deviance from 0 or 1 in read frequency (those loci that on average deviated more than 1%), which removed 2.8% of the loci. Finally, genotypes were called per individual if a locus had a coverage higher than 15 and the alternative allele frequency of either AfIR974 or AfIR964 was 90% or higher.

Further analyses of genetic recombination were performed using the R/qtl2 package, detailed in the supplementary code material.

### Genetic mapping of acriflavine resistance

Acriflavine resistance was determined by plating approximately 10,000 spores in a 10 µL droplet on Complete Media with 50 µg/L acriflavine. Droplets were incubated for 3 days at 37°C before phenotyping. Growth/no growth was scored as a binary trait, and QTL mapping was performed with R/qtl2 using the scanone function to detect the responsible QTL [76]. Genes were visualized using the gggenes package (v0.41) (https://wilkox.org/gggenes/), based on the GFF annotation of the AfIR974 parent.

### *cyp51A* Intragenic recombination

To assess recombination within *cyp51A*, the strains SPF94 (AfS35 cyp51A$^{TR34}$::*hph*) and SPF96 (AfS35 cyp51A$^{L98H}$::*hph*) were each crossed with the wild-type strain AfIR974 on Compost Agar media (60 g/L freeze dried mushroom compost, 20 g/L agar). After incubation for 8 weeks, mature cleistothecia were harvested and heatshocked as above. Colonies resistant to both 4 µg/mL itraconazole as well as 150 µg/mL hygromycin (Ducha Chemicals) were isolated and mating type was determined using previously described primers [12]. Offspring of compatible mating types with either TR$_{34}$ or L98H variants were then crossed (94–4: TR$_{34}$ MAT1-1; 96–6: L98H MAT1-2). From this cross single cleistothecia were isolated and heatshocked in 500 µL H$_2$O+0.05% Tween-80. From this ascospore suspension, 4 aliquots of 3 µL were used for count plating on Malt Extract Agar (30 g/L malt extract, 15 g/L agar), and the remaining plated on large (16 cm diameter) petri plates with Malt Extract Agar, either with or without 10 µg/mL itraconazole. Count plates were incubated for 36 h, and itraconazole plates were incubated 72 h. To confirm the double recombinant offspring, isolated colonies from the itraconazole plates were subcultured onto MEA + 4 µg/mL itraconazole. Spores were harvested and crude genomic DNA was extracted [77]. We then PCR amplified the *cyp51A* region using forward primer P-A7 [78] and reverse primer CYP51A_R119 (5′-CACGTCCGATCCGAAAA CGGGG-3′), followed by Sanger Sequencing.

### Supporting information

**S1 Fig. Synteny of two parental strains and reference strain Af293. (A)** Dotplot comparison of sequence similarity using minimap2 of AfIR974 assembly against reference Af293 assembly. **(B)** Comparison of AfIR964 against Af293. **(C)** Comparison of assemblies of 2 parental strains AfIR964 and AfIR974. Data underling this figure can be found at https://doi.org/10.5281/ zenodo.8167717.
(DOCX)

**S2 Fig. Comparison of physical and genetic distance between markers. (A)** Comparison of raw dataset, including presumed gene conversions. Each data point indicates the distance between a pair of adjacent markers, with physical distance shown on a log scale. Density of markers shown by curve for both physical and genetic distances. Note that appreciable recombination between markers is only seen above 100 bp. **(B)** Comparison of dataset after removal of gene conversions. Data underling this figure can be found at https://doi.org/10.5281/zenodo.8167717.
(DOCX)

**S3 Fig. Comparison of cleaned dataset against uniformly distributed simulated maps. (A–I)** Plots of genome-wide observed distances between crossovers (dotted line) and simulated distances between crossovers (solid line). Rows indicate simulations for different genetic map lengths as indicated. Right panel show the same data as the left panel but using a log scale on the x-axis. Data underling this figure can be found at https://doi.org/10.5281/zenodo.8167717.
(DOCX)

**S4 Fig. Comparison of genetic map of *A. fumigatus* to Stapley and colleagues dataset. (A)** Small dots indicate values extracted from Stapley and colleagues comparing genome size to genetic map length. Larger triangle indicates *A. fumigatus*. **(B)** Similar to A except controlling for chromosome number. Data underling this figure can be found at https://doi.org/10.5281/zenodo.8167717.
(DOCX)

**S5 Fig. Recombination landscapes of all 8 chromosomes of *A. fumigatus*.** Plots are similar to Fig 2B of the main text but shown for all 8 chromosomes. Dotted line indicates the genome-wide average recombination rate, solid line indicates recombination rate across 50 kb windows. Small vertical hash lines at top of plots indicate marker positions, and large circle indicates estimated centromere position. Data underling this figure can be found at https://doi.org/10.5281/zenodo.8167717.
(DOCX)

**S6 Fig. Intragenic *cyp51A* recombination in *A. fumigatus* ascospore progeny. (A)** Whole contents of single cleistothecia plated on Malt Extract Agar plates. **(B)** Whole contents of a single cleistothecium selected for *cyp51A* recombinants by plating instead on MEA + 10 µg/mL itraconazole. Bottom right plate is the selection plate shown in Fig 4.
(DOCX)

**S1 Table. Parental genome assembly statistics.** Data underling this figure can be found at https://doi.org/10.5281/zenodo.8167717.
(DOCX)

**S2 Table. Types of variants and their effects.** Data underling this figure can be found at https://doi.org/10.5281/zenodo.8167717.
(DOCX)

**S3 Table. Genetic map length statistics.** From left to right, chromosome, chromosome length, number of variants detected, recombination events per offspring, raw recombination fraction, rarefied recombination fraction, and gene conversion (GC) corrected map length is shown. Data underling this figure can be found at https://doi.org/10.5281/zenodo.8167717.
(DOCX)

**S4 Table. Details of crosses used for recombination analyses.** All parental isolates are azole sensitive Dutch or Irish environmental isolates unless specified otherwise.
(DOCX)

## Acknowledgments

We thank Dr. Eugene Gladyshev and Dr. Raphael Mercier for input on a previous version of this manuscript. We gratefully acknowledge Dr. W.M. Moye-Rowley for sharing the L98H and TR$_{34}$ single mutants used in the crosses for Fig 4. We thank Dr. Paul Verweij for sharing the AfIR964 and AfIR974 parental strains. Dr. Erik Wijnker as well as other members of the Laboratory of Genetics provided valuable feedback during the writing process.

## Author Contributions

**Conceptualization:** Alfons J. M. Debets, Eveline Snelders.

**Formal analysis:** Ben Auxier, Joost van den Heuvel.

**Funding acquisition:** Eveline Snelders.

**Investigation:** Ben Auxier, Joost van den Heuvel, Eveline Snelders.

**Methodology:** Ben Auxier, Alfons J. M. Debets, Felicia Adelina Stanford, Johanna Rhodes, Frank M. Becker, Francisca Reyes Marquez, Reindert Nijland, Paul S. Dyer, Matthew C. Fisher, Joost van den Heuvel, Eveline Snelders.

**Software:** Ben Auxier, Joost van den Heuvel.

**Supervision:** Eveline Snelders.

**Visualization:** Ben Auxier, Joost van den Heuvel.

**Writing – original draft:** Ben Auxier.

**Writing – review & editing:** Ben Auxier, Alfons J. M. Debets, Joost van den Heuvel, Eveline Snelders.

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
