## [Editor Report · Decision Letter 0]

2 May 2023

Dear Dr. Snelders, 

Thank you for submitting your manuscript entitled "Meiosis in the human fungal pathogen Aspergillus fumigatus can produce the highest known number of crossovers" for consideration as a Research Article by PLOS Biology.

Your manuscript has now been evaluated by the PLOS Biology editorial staff, as well as by an academic editor with relevant expertise, and I am writing to let you know that we would like to send your submission out for external peer review.

Once your full submission is complete, your paper will undergo a series of checks in preparation for peer review. After your manuscript has passed the checks it will be sent out for review. To provide the metadata for your submission, please Login to Editorial Manager (https://www.editorialmanager.com/pbiology) within two working days, i.e. by May 04 2023 11:59PM.

***If your manuscript has been previously peer-reviewed at another journal, PLOS Biology is willing to work with those reviews in order to avoid re-starting the process. Submission of the previous reviews is entirely optional and our ability to use them effectively will depend on the willingness of the previous journal to confirm the content of the reports and share the reviewer identities. Please note that we reserve the right to invite additional reviewers if we consider that additional/independent reviewers are needed, although we aim to avoid this as far as possible. In our experience, working with previous reviews does save time. 

Kind regards,

Paula

---

Senior Editor

PLOS Biology

---

## [Decision Letter · Decision Letter 1]

23 Jun 2023

Dear Dr. Snelders,

Thank you for your patience while your manuscript "Meiosis in the human fungal pathogen Aspergillus fumigatus can produce the highest known number of crossovers" went through peer-review at PLOS Biology. Your manuscript has now been evaluated by the PLOS Biology editors, an Academic Editor with relevant expertise, and by several independent reviewers.

In light of the reviews, which you will find at the end of this email, we are pleased to offer you the opportunity to address the comments from the reviewers in a revision that we anticipate should not take you very long. We will then assess your revised manuscript and your response to the reviewers' comments with our Academic Editor aiming to avoid further rounds of peer-review, although might need to consult with the reviewers, depending on the nature of the revisions. We think that the manuscript should be a Short Report, please check this option when sending your revision.

Short Reports present the results from a limited set of experiments that can generally be summarized in 3-4 figures or fewer. The outcomes should be self contained, rather than fitting within the narrative arc of a larger research project or article.

Please also address the following policy and formatting requests:

**1. DATA POLICY:**

A) Supplementary files (e.g., excel). Please ensure that all data files are uploaded as 'Supporting Information' and are invariably referred to (in the manuscript, figure legends, and the Description field when uploading your files) using the following format verbatim: S1 Data, S2 Data, etc. Multiple panels of a single or even several figures can be included as multiple sheets in one excel file that is saved using exactly the following convention: S1_Data.xlsx (using an underscore).

B) Deposition in a publicly available repository. Please also provide the accession code or a reviewer link so that we may view your data before publication.

Regardless of the method selected, please ensure that you provide the individual numerical values that underlie the summary data displayed in the following figure panels as they are essential for readers to assess your analysis and to reproduce it: Figures 1ABCDEFG, 2ABCDEF, 3BC, 4B, Supplementary Figures S1ABC, S2AB, S3ABCDEFGHI, S4AB, S5.

**Please also ensure that figure legends in your manuscript include information on where the underlying data can be found, and ensure your supplemental data file/s has a legend.**

**2.** Please note that sole deposition of data or code to GitHub would not be compliant with our policies, as this could be changed after publication (https://journals.plos.org/plosbiology/s/data-availability). However, once the data/code is final, you can archive your publicly available GitHub data to Zenodo. Once you do this, it will also generate a DOI number that you can provide us with. See the process for doing this here: https://docs.github.com/en/repositories/archiving-a-github-repository/referencing-and-citingcontent

**3. **We suggest a change in the title: "The human fungal pathogen Aspergillus fumigatus can produce the highest known number of meiotic crossovers".

**IMPORTANT - SUBMITTING YOUR REVISION**

*Resubmission Checklist*

*Published Peer Review*

*PLOS Data Policy*

*Blot and Gel Data Policy*

Sincerely,

Paula

---

Senior Editor

PLOS Biology

REVIEWS:

Reviewer #1: Genome dynamics.

Reviewer #2: Chromosomes, mitosis and meiosis.

Reviewer #3: Meiosis recombination.

Reviewer #1: Auxier et al. characterized meiotic recombination events genome wide in the progeny of four Aspergilus fumigatus intraspecific hybrids. In two of them, they found the highest meiotic crossover frequency ever recorded, in agreement with the very short LD50 previously described in this species. This property is not shared with the two other hybrids studied, indicating a high variability in crossover frequency within this species. The authors further took advantage of this elevated meiotic crossover frequency for precise QTL mapping, and to generate intragenic recombined offspring at the cyp51A locus.

Given the novelty of their main finding, the elevated recombination frequency of A. fumigatus, the authors took great care of challenging it with stringent criteria to make sure it was not artifactual (ie coming for gene conversion events not associated with any crossover).

Overall, the results are convincing and bring new and interesting light on the variability of the regulation of meiotic recombination, identifying a new "world champion" with higher crossover number per chromosome than the former champions that are Schizosaccharomyces pombe and Saccharomyces cerevisiae.

Comments

1. The introduction is so minimal that it is even shorter than the abstract that contains more introductive information: this cannot stand as it is. The authors should provide a review of the literature about Aspergilus fumigatus meiotic recombination.

2. Unlike in budding and fission yeasts where the four gametes per meiosis can be easily isolated for genotyping, here a main limitation to the analysis is that it relies on random isolation of haploid offspring. Therefore, crossover identification relies on only half of the recombined material, the other complementary /reciprocal half being unavailable for genotyping. I would like the authors to consider this as a possible caveat to this strategy since crossovers are not directly seen but inferred.

I acknowledge that I am not a specialist of A. fumigatus meiosis. From what I understand, gamete formation takes 4-6 weeks. Is it known for sure that during this time, only one meiosis takes place during gamete formation? What if several meioses take place during this time in combination with spore mating? Can this possibility be rule out? Alternatively, in budding yeast the mechanism of return to growth (RTG) has been described and consists of an abortive meiosis after Spo11-DSBs are formed. In case several rounds of RTG take place prior full meiosis completion, the number of crossover sites determined from the haploid offspring will look artificially high compared to the actual number of crossover sites per meiosis.

3. Given the SNP density, simple gene conversions not associated to any crossover (ie non-crossovers) should be readily identifiable. Although somewhat mentioned along the text, it looks to me that a specific paragraph should be dedicated to the frequency of non-crossovers, their putative properties (length etc).

Furtheremore, the definition of non-crossover / gene conversion should be revised. Gene conversion result from the repair of mismatches that are formed in any recombination event, whether it gives rise to a crossover or a non-crossover.

4. Unsean double CO (CO = crossover). Although heterogenous, the marker density is such that it leaves very little room for unsean double CO.

5. The reference for non-crossover / gene conversion in A. fumigatus is Chen et al 2007 which is not specific to A. fumigatus. What is known about gene conversions and gene conversion tract length in A. fumigatus?

6. Some clarification is needed in the comparison of the recombination properties of A. fumigatus with respect to published data and reference species such as S. cerevisiae.

Considering 132 CO per haploid offspring, it is expected to see about 264 CO per meiosis in A. fumigatus, ie 264 CO / 30 Mb ie 8.8 CO per Mb.

In S. cerevisiae, Mancera et al. reported ca. 90 CO per meiosis, ie 90 CO / 12 Mb ie 7.5 CO per Mb.

In conclusion, the number of CO per meiosis is clearly higher in A. fumigatus than is S. cerevisiae, but the number of CO per Mb is comparable between the two species. Given the strict correlation between chromosome size and CO number in S. cerevisiae, it looks like the same applies to A. fumigatus, but with a higher total CO number per meiosis resulting from a longer genome.

7. Figure 2F. The axes annotations do not match with the legend of the figure. Double check.

8. CO interference:

Although S. pombe is considered as lacking CO interference, work from G. Smith lab revealed some short-range CO interference likely resulting from DSB spacing, controlled by Tel1.

Interestingly, J. Fung lab revealed that the absence of Tel1 also impacts on CO interference in S. cerevisiae, suggesting part of CO interference in S. cerevisiae results from DSB spacing.

Finally, recent work done in the budding yeast Lachancea waltii revealed the existence of a weak short-range interference independent of the ZMM CO interference pathway.

In conclusion, I suggest the authors double check their analysis of CO interference to specifically determine if they also detect such a short-range CO interference which would also likely result from the DSB spacing.

9. Figure 2B and S5.

In most species studied so far, CO are suppressed around centromeres. Here, chromosome 1 is used as a representative chromosome in Fig 1B, but it is the only chromosome not showing a CO depletion around the centromere, while all the seven other chromosomes show a depletion around it.

I therefore suggest using another chromosome for Fig 2B in order to be as representative as possible.

However, I am surprised to see a CO coldspot on chromosome 1 ca 0.7MB right of the centromere. I suggest the authors double check for the precise annotation of the centromere of chromosome 1, and whether it is annotated at the right place.

Typo:

Is occurring "in" azole-contaminated bulb waste heaps

Reviewer #2: This work from Auxier et al presents interesting and useful observations that should be of interest to the readers of PLoS Biology.

I have only two comments.

1. Title: ...."the highest known number of crossovers". The highest known number of crossovers per what? Per Mb? Per chromosome? Per genome? From the abstract it seems that the authors are interested in the number per chromosome (per pair of homologs). But the text emphasizes the number of crossovers per total genome size in Mb (Figure 1F) and the number of crossovers per Mb per chromosome (Figure 1G). The authors seem to be mostly interested in the genetic consequences of crossing-over, in which case crossovers per Mb is most directly relevant and the title should emphasize this feature. The absolute number of crossovers per chromosome is of interest if one is interested in the role of crossing-over in meiotic chromosome segregation, but this is not the authors' focus. The number per genome is less interesting without reference to total genome size and/or total chromosome number.

2. The authors understanding of the mechanics of meiosis should be improved.

- "To some extent, the high numbers of crossovers that we observe may be explained as the lack of the SC required to ensure that at least one crossover occurs per chromosome pair, necessary for proper meiotic chromosome reduction (Egel-Mitani et al., 1982; Snow, 1979)."

The two references cited are from 1979 and 1982. A lot has happened since then. In two other fungi (budding yeast and Sordaria macrospora) SC is not required, mechanistically, to ensure that at least one crossover occurs per chromosome pair and, in both cases, is the downstream outcome of the processes that determine crossover numbers and interference.

- "To some extent, the high numbers of crossovers that we observe may be explained as the lack of the SC required to ensure that at least one crossover occurs per chromosome pair, necessary for proper meiotic chromosome reduction (Egel-Mitani et al., 1982; Snow, 1979)." The authors might want to read the Arabidopsis literature which identifies genetic situations in which the numbers of crossovers are dramatically increased, and those crossovers don't exhibit interference. Perhaps Aspergillus fumigatus is an analogous mutant case, particularly since the "two strains" in the cross are not genetically identical, so the high level of crossovers could be a consequence of genetic heterozygosity (irrespective of structural variations). Ref: Serra H, Lambing C, Griffin CH, Topp SD, Nageswaran DC, Underwood CJ, Ziolkowski PA, Séguéla-Arnaud M, Fernandes JB, Mercier R, Henderson IR. Proc Natl Acad Sci U S A. 2018.

Reviewer #3: This is truly an excellent paper. The authors have been extremely careful in their analysis. I was especially impressed by the manner in which they coped with the complexity of gene conversion. I was initially a bit worried about an absence of a perspective of 'meiotic biology - specifically, the lack of interference and the absence of the SC'. But the authors addressed these issues beautifully, along with the appropriate comparison to S pombe, on page 6. My one suggestion would be to encourage the authors to make reference of this discussion of meiotic biology in their Abstract. It is important.

The paper is a lovely contribution to several fields and has tangible application to public health.

This a very good paper and I strongly encourage publication. I feel badly that I could not find any significant flaw. I tried. But it is a really nice paper!

---

## [Editor Report · Decision Letter 2]

27 Jul 2023

Dear Dr Snelders,

Thank you for the submission of your revised Short Report "The human fungal pathogen Aspergillus fumigatus can produce the highest known number of meiotic crossovers" for publication in PLOS Biology. On behalf of my colleagues and the Academic Editor, Joseph Heitman, I am pleased to say that we can in principle accept your manuscript for publication, provided you address any remaining formatting and reporting issues. These will be detailed in an email you should receive within 2-3 business days from our colleagues in the journal operations team; no action is required from you until then. Please note that we will not be able to formally accept your manuscript and schedule it for publication until you have completed any requested changes.

PRESS

Sincerely, 

Paula

---

Senior Editor

PLOS Biology
